

# Trapping of HCl and oxidized, organic trace-gases in growing ice at temperatures relevant for cirrus clouds.

Matthias Kippenberger[1], Gerhard Schuster[1], Jos Lelieveld[1] and John N. Crowley[1]

[1]Max-Planck-Institut für Chemie, Division of Atmospheric Chemistry, Mainz, 55128 Germany.

*Correspondence to*: John Crowley: (John.crowley@mpic.de)

**Abstract.** The uptake of hydrochloric acid (HCl), ethanol ($C_2H_5OH$), 1-butanol (1-$C_4H_9OH$), formic acid $HC(O)OH$ and tri-fluoro acetic ($CF_3C(O)OH$) acid to growing ice surfaces was investigated at temperatures between 194 and 228 K. HCl displayed extensive, continuous uptake during ice growth, which was strongly dependent on the ice growth velocity, the temperature of the ice surface and the gas phase concentration of HCl. Tri-fluoro acetic acid was also observed to be trapped

in growing ice, albeit approximately an order of magnitude less efficiently than HCl, whereas the adsorption and desorption kinetics of ethanol, 1-butanol, formic acid on ice was not measurably different to that for non-growing ice, even at very large ice-growth rates. We present a parameterisation of the uptake coefficient for HCl on growing ice films ($\gamma_{trap}$) and compare the results to an existing framework that describes the non-equilibrium trapping of trace gases on ice. The trapping of HCl in growing ice crystals in the atmosphere is assessed and compared to the gas- and ice-phase partitioning resulting from

equilibrium surface adsorption and solubility.

## 1 Introduction

Ice clouds, which represent a surface area that exceeds that of the underlying earth by about an order of magnitude, have a significant influence on the chemical composition of the atmospheric gas and condensed phases. They can reversibly remove trace-gases by physical adsorption (non-reactive uptake), or provide a surface to catalyse heterogeneous processes which are

otherwise too slow to be of importance for the atmosphere. Examples of physical adsorption are the uptake of $HNO_3$ and $H_2O_2$ to cirrus clouds, which can result in the re-partitioning from the gas to the particle phase and to a vertical redistribution of the trace gases for example by evaporating hydrometeors during gravitational sedimentation (Popp et al., 2004; von Kuhlmann and Lawrence, 2006; Popp et al., 2007; Marécal et al., 2010; Pouvesle et al., 2010). The best known example of reactive uptake is the interaction between HCl and $ClONO_2$ on ice particles, which plays a central role in polar ozone

destruction in the lower stratosphere (Solomon et al., 1986; Molina et al., 1987).

The nonreactive uptake of trace gases to atmospheric ice particles is often described using Langmuir adsorption isotherms. Whilst there is a priori no reason why the Langmuir isotherm should be applicable to trace gas uptake to an ice surface, which at atmospheric temperatures is extremely dynamic (Girardet and Toubin, 2001), it can be used to parameterise experimental observations for a large variety of atmospheric trace gases reasonably well (Abbatt, 2003; Crowley et al.,





2010). Further, for substances which hydrogen bond to the ice surface, the Langmuir partition coefficient can be related to the free energy of condensation (Sokolov and Abbatt, 2002; Pouvesle et al., 2010), though this relationship breaks down for strong acids such as $HNO_3$ and HCl. Due to their important roles in atmospheric chemistry, the interaction of HCl and $HNO_3$ with ice surfaces has been studied intensively over the past three decades. With very few exceptions, the vast majority of these studies, reviewed by IUPAC (Crowley et al., 2010), were performed under equilibrium conditions where the ice neither grows nor evaporates. This situation does not always apply to atmospheric ice which frequently experiences sub- and super-saturation (Gao et al., 2004; Lee et al., 2004; Popp et al., 2007; Comstock et al., 2008) resulting in episodes of net evaporation and growth, respectively.

A trace gas colliding with a growing ice surface may be considered to have a finite probability of being "trapped" in the advancing ice crystal before it can desorb, effectively removing it from the gas-phase until the ice particle evaporates. Trapping can thus change the gas-to-particle partitioning of a trace gas and, as the trace-gas locates into (not onto) the particle, it may not be readily available for heterogeneous reactions with other trace gases colliding with the ice surface.

Evidence for the efficient trapping of a trace gas in growing ice crystals is provided by analysis of results from several airborne field campaigns, which, by measurement of gas- and particle-phase nitrate led to the conclusion that the uptake of $HNO_3$ is efficiently enhanced during ice growth events (Kärcher and Voigt, 2006). However, the inherent variability in atmospheric conditions (temperature, ice growth rates and nitric acid concentrations) complicates the analysis of such data sets. To aid inclusion of the trapping of trace gases into atmospheric and climate models Kärcher and co-workers (Kärcher and Basko, 2004; Kärcher et al., 2009) have developed a framework ("trapping model") that considers the surface-physical processes involved in transferring a molecule colliding with the ice surface to the bulk. The trapping model was developed in order to understand aircraft-based observations of large $HNO_3$ concentrations in ice crystals. The validation of this semi-empirical model and especially its extrapolation to other trace gases has to date not been possible owing to a lack of laboratory experimental data under controlled conditions and covering sufficient parameter space.

We present a detailed laboratory study that investigates the trapping efficiency for several different atmospheric trace gases under varying conditions of ice surface temperature $T$, ice growth velocity $V_{IG}$, and HCl concentration [HCl].

## 2 Experimental

The experiments were conducted using a newly designed apparatus that enables generation of a well-defined water super-saturation during trace gas uptake to an ice surface growing at a constant, known rate. The apparatus (Fig. 1) consists of three chambers made of stainless steel. The cylindrical, lower chamber containing the ice sample ("ice-chamber", diameter = 4.86 cm, height = 1.3 cm) is isolated from the "upper chamber" (volume = 155 cm³) by a moveable quartz plate. The bottom of the ice chamber and the lowest 4 mm of its walls are in thermal contact with a copper block, which is cooled by a flow of ethanol from a cryostat. A thermocouple, located just under the ice surface is used to obtain the approximate temperature of the ice. To reduce heat-transfer to the ice and ensure a sharp temperature gradient between the reactor walls in contact with





the ice and the rest of the reactor, the cylindrical parts of the ice-chamber were constructed from very thin stainless steel (0.4 mm). Additionally, the upper chamber and the upper section of the walls of the ice-chamber were heated resistively to keep them at room-temperature. The upper chamber hosts the gas inlets through which Helium, water and the trace gas of interest are introduced as well as a pressure gauge (10 Torr, MKS Baratron, 1Torr = 1.333 mbar) and the connection to a quadrupole

mass spectrometer (QMS, see below). Interaction between trace gas and ice surface is achieved when the quartz plate is withdrawn into a third chamber, which is continuously flushed with 100 cm$^3$ (STP) min$^{-1}$ (sccm) of Helium during experiments.

All experiments reported here were conducted at a total pressure of 1.33 mbar and at ice surface temperatures between 194 and 227 K. The gas phase temperature was 298 K during all experiments. The flow rate of He was between 700 and 800

sccm. The walls of upper and lower chambers were coated with very thin Teflon films (Chemours, FEPD, 121).

### 2.1 Preparation and characterization of the ice surface

Ice was prepared by freezing 0.6-1.2 ml of liquid water (HPLC grade, Merck) in the ice compartment at 243 K and subsequently cooling it down to the temperature of the experiment. The ice substrate had a geometric surface area of 18.6 cm$^2$, a thickness of 1-2 mm and appeared translucent and smooth from visual inspection. Occasionally some cracks formed

during cooling, presumably due to thermal stress in the relatively thick ice. The temperature at the ice surface was calculated from the vapour pressure of water at which net evaporation or condensation was zero. This was achieved in an iterative procedure in which the water vapour concentration (monitored by the QMS in electron impact mode, see below) was adjusted via a flow controller until the concentration no longer changed upon opening and isolating the ice-chamber. The QMS sensitivity to water vapour was determined by flowing a known amount of pure H$_2$O through a calibrated mass flow

controller and diluting this with an accurately known He-flow before passing the combined flow through the upper chamber (with the ice-chamber closed). Subsequently, the measured water-vapour pressure above the ice was then converted to the ice-surface temperature using literature vapour pressure data (Wagner, 1994). The ice-surface temperature was generally higher than that measured by the thermocouple in the bottom plate of the ice chamber, the largest difference ($\approx$ 5 K) found at the lowest temperatures and when using the thickest ice films. At 235 K, the difference was $\approx$ 0.5 K for thin films and 1.5 K

for thick films. However, the temperature registered by the thermocouple is only used as a guide to the ice-surface temperature and this observation has no impact on our results or analysis. The uncertainty in the ice-surface temperature measured via the water vapour concentration is $\pm$ 2 K.

The net H$_2$O-flux to the ice surface during periods of super-saturation, $J_{H2O}$, was calculated from the difference in the calibrated mass signal at $m/z$ 20 with the ice-chamber isolated ("without ice") and open ("with ice") at different water vapour

concentrations. The difference in signal intensity was assumed to arise solely from the uptake of H$_2$O molecules by the ice surface. The ice growth velocity ($V_{IG}$, ms$^{-1}$) was derived from the measured flux of H$_2$O molecules from the gas phase onto the ice surface and assuming an ice-density of 0.92 g cm$^{-3}$. The growth velocities were found to increase linearly with ice super saturation (i.e. water vapour concentration in the reactor) in the range relevant to the growth experiments. The



calibration was also periodically performed on HCl-covered ice and ice that had grown for several hours in an atmosphere containing HCl at normal experimental concentrations. The deviation between the different calibrations was usually less than 1%, indicating that the presence of HCl did not alter the growth velocity of the ice significantly. Ice-growth velocities during the uptake experiments were between $1.1 \times 10^{-9}$ and $8.0 \times 10^{-8}$ m s$^{-1}$, which is typical of cirrus clouds with ice super
saturations between 1% and 20% (Bailey and Hallett, 2012).

## 2.2 Detection of trace gases

All gases were detected online using a quadrupole mass spectrometer (Balzers QMA 410) which was coupled to the ice/trace-gas interaction chamber via a glass tube coated with Teflon (Chemours, FEPD, 121). Details of the mass-spectrometer, which can use either electron impact or chemical ionisation are given in Winkler et al. (2002) and von
Hessberg et al. (2008). Water ($H_2^{18}O^+$, $m/z$ 20), ethanol and 1-butanol (both $CH_3O^+$, $m/z$ 31) and tri-fluoro acetic acid ($CF_3^+$, $m/z$ 69) were detected using electron impact, HCl and HC(O)OH were detected in chemical ionisation mode using $SF_6^-$ ions. The latter were formed by passing a 0.2 % mixture of $SF_6$ in $N_2$ (Air liquid) through a $^{210}Po$ radioactive source (Model P-2021-1000, NRD). $SF_6^-$ reacts with HC(O)OH and HCl to form product ions at m/z 65 (HC(O)OHF$^-$) and 162 ($SF_5Cl^-$) (Huey et al., 1995; Lovejoy and Wilson, 1998).

The detection limits (based on the background noise levels at the respective signals) in electron impact mode for the organic trace-gases were calculated to be in the range 7.5 to $9.5 \times 10^9$ molecule cm$^{-3}$ (0.17 s signal integration). For chemical ionisation, the detection limits for HC(O)OH and HCl were $1.9 \times 10^9$ molecule cm$^{-3}$ (0.30 s) and $3.2 \times 10^9$ molecule cm$^{-3}$ (0.17 s), respectively. The mass spectrometric signals were calibrated by monitoring the pressure drop in a calibrated volume which contained a manometrically prepared mixture of the trace gas of interest in He 5.0 (Westfalen). An HCl mixture of
3.95 vol% in He was used as stock supply (Air Liquide) and further diluted to mixing-ratios of $\approx 2 \times 10^{-4}$. The concentration of the stock mixture was determined by passing a known amount through two wash bottles, each containing 250 ml of pure water. The resulting aqueous solutions were then titrated with 0.01 N volumetric standard solution of sodium hydroxide in water (Carl Roth) with phenolphtalein as indicator. The standard deviation of different titrations was 0.1 to 2% between runs. Owing to diffusion limitations, the excess water vapour during super-saturation does not completely condense on the ice.
This results in an increase in gas-phase $H_2O$ reaching the detector during phases of ice growth, which resulted in an increase in QMS sensitivity to HCl (usually less than 10 %). The dependence of the HCl-detection scheme on [$H_2O$] was linear and was taken into account when converting QMS signals to HCl concentrations during ice growth.

## 3 Results and discussion

Whilst experiments were conducted for all of the trace-gases listed above, only HCl and, to a lesser extent, TFA displayed
evidence for trapping in growing ice. We therefore first present a typical dataset for HCl to illustrate the data analysis.





## 3.1 HCl uptake

A representative dataset showing the uptake of HCl onto a growing ice surface is displayed in Fig. 2. Initially, HCl is introduced into the reactor with the quartz-plate closed and a stable concentration was established (t < 150 s). A t = 150 s the quartz-plate was withdrawn, exposing the fresh ice surface to HCl. The concentration of HCl drops immediately as HCl

adsorbs to the fresh ice surface and then gradually returns towards its initial value as the surface saturates. This first uptake takes place at ice-water vapour equilibrium (super-saturation = 0%). At t ~ 1040 s a super-saturation of 41% was created by increasing the water concentration in the reactor, resulting in growth of the ice surface at 2 nm s$^{-1}$. The permanent decrease of the HCl gas-phase concentration during this ice-growth phase is evidence for continuous uptake (and thus trapping) of HCl. At t ~ 1200 s and 1320 s further increases in the ice-super-saturation (to 82 and 122%, respectively) leads to enhanced

continuous uptake. At t ~ 1520 s the equilibrium conditions were restored (super-saturation of 0%) and the HCl signal returns to its initial value. The effective uptake coefficient during trapping ($\gamma_{\text{trap}}$), was calculated using:

$$\gamma_{\text{trap}} = \frac{d[\text{HCl}]}{dt_{ice}[\text{HCl}]} \frac{4V}{A_{ice}\bar{c}} \qquad (1)$$

Where $V$ and $A_{\text{ice}}$ are the volume of the reactor ($\approx$ 161.3 cm$^3$) and the ice surface area, respectively and $\bar{c}$ is the mean molecular velocity of HCl at 298 K ($\approx$ 42000 cm s$^{-1}$). $t_{\text{ice}}$ refers to the time the HCl is in contact with the ice surface and

varied between $\approx$ 18 and 21 ms, depending on the total flow. The right-hand, y-axis indicates the flux of HCl (molecules s$^{-1}$) passing through the reactor. The net flux of HCl ($J_{\text{HCl}}$) to the surface is calculated from the difference in signal when the HCl exposure to the ice is modulated by withdrawing the sliding isolator.

$$J_{\text{HCl}} = F_{\text{HCl}} - F_{\text{HCl}}^{\text{ice}} \qquad (2)$$

The molar fraction of HCl in the growing ice surface, $M_{\text{HCl}}$, is calculated from the relative flux of HCl and condensing water:

$M_{\text{HCl}} = J_{\text{HCl}} / J_{\text{H2O}}$          (3)

Datasets such as those exemplified in Fig. 2 were obtained for a number of temperatures ($T$, varied between 194 and 227 K), HCl concentrations ([HCl], varied between 6.4 × 10$^9$ and 2.2 × 10$^{11}$ cm$^{-3}$) and ice growth velocities ($V_{\text{IG}}$, varied between 1 and 68 nm s$^{-1}$).

Figure 3b shows the variation of $\gamma_{\text{trap}}$ with the growth velocity ($V_{\text{IG}}$) at two different HCl concentrations (2.3 × 10$^{10}$ and 7.1 ×

10$^{10}$ molecule cm$^{-3}$) and an ice temperature of 210 ± 1 K. In each case, $\gamma_{\text{trap}}$ depends linearly on the growth velocity of the ice surface. As ice-growth was initiated only after HCl had returned to its concentration prior to exposure (i.e. when $\gamma_{\text{trap}} = 0$) we expect a direct proportionality between $V_{\text{IG}}$ and $\gamma_{\text{trap}}$. This was observed in most experiments, with the intercept in plots such as those in Fig. 3, deviating insignificantly from zero. In some cases, however, the deviation of the intercept from zero was significant and relates to an offset in $V_{\text{IG}}$, which was calculated relative to the mass-spectrometer signal before HCl exposure

began (e.g. at time = zero in Fig. 1). A drift in mass spectrometer sensitivity or the ice temperature between time = zero and the onset of H$_2$O-supersaturation (often > 1000 s later) may shift the plot of $\gamma_{\text{trap}}$ versus $V_{\text{IG}}$ but should not impact on values of $d\gamma_{\text{trap}} / dV_{\text{IG}}$ which were obtained in quick succession and therefore less susceptible to drifts.





A linear relationship between $\gamma_{trap}$ and $V_{IG}$ was observed for values of $\gamma_{trap}$ up to ~ 0.20. At larger uptake coefficients (e.g. at larger ice growth velocities) the uptake of HCl to the ice is increasingly influenced by effects of diffusive transport, whereby gradients in the HCl concentration close to the ice-surface result in a reduction in the slope. As no simple scheme for correction of the diffusive limitation to uptake exists for the complex geometry of the reactor, values of $\gamma_{trap}$ greater than

0.018 were neglected in the final analysis.

In Fig. 3a, taking the same dataset, we plot the flux of HCl ($J_{HCl}$ in molecules s$^{-1}$) to the surface during ice growth. $J_{HCl}$ is calculated from the difference between the flux observed by the mass-spectrometer with and without interaction between HCl and the ice surface, as illustrated in Fig. 2. In contrast to $\gamma_{trap}$, the two sets of measurements of $J_{HCl}$ at different [HCl] show the same dependence on $V_{IG}$. As $\gamma_{trap}$ represents the efficiency of HCl removal from the gas-phase on a per-collision

basis and $J_{HCl}$ is the absolute number of HCl lost, we can conclude that, in our HCl concentration range, the trapping process is not limited by the collision rate of HCl with the ice surface, or the net absorption rate, the same absolute flux being observed at a given ice growth rate for both high and low HCl concentrations and thus high and low collision / adsorption rates.

Returning to Fig. 3b we see that, for a given ice growth velocity, the trapping efficiency depends on the [HCl] concentration.

For example, an ice growth velocity of 10 nm s$^{-1}$, results in values of $\gamma_{trap}$ that increase by a factor of four when [HCl] is reduced from $7.1 \times 10^{10}$ molecule cm$^{-3}$ to $2.3 \times 10^{10}$ molecule cm$^{-3}$. The HCl concentration can be converted to a fractional surface coverage $\theta_{HCl}$ (i.e. the equilibrium coverage which would have prevailed if the ice were not supersaturated) according to the Langmuir adsorption model:

$$\theta_{HCl} = \frac{N}{N_{max}} = \frac{K_{Lang}[HCl]}{K_{Lang}[HCl]+1} \qquad (4)$$

where $K_{Lang}$ is the Langmuir constant that describes the partitioning between molecules in the gas phase and those adsorbed on the surface; $N_{max}$ is the maximum surface coverage at equilibrium. We converted our gas-phase HCl concentrations into surface coverages that would have been obtained under equilibrium conditions using the latest evaluation of adsorption isotherms (IUPAC, 2019) with $K_{Lang} = 9.6 \times 10^{-11}$ cm$^3$ molec$^{-1}$, independent of temperature in the range of our experiments. We recognise that the calculated values of $\theta_{HCl}$ during ice-growth do not reflect actual surface-coverages, but find this a

more useful parameter than the HCl concentration when interpreting and parameterising the values of $\gamma_{trap}$ obtained during growth (see below). Fractional surface coverages are also used in the trapping model (Kärcher and Basko, 2004; Kärcher et al., 2009), which we return to later.

Figure 4b presents the dependence of $\gamma_{trap}$ on $V_{IG}$ at three different ice temperatures, 195 K (with [HCl] = $2.2 \times 10^{10}$ molecule cm$^{-3}$), 211 K (with [HCl] = $2.5 \times 10^{10}$ molecule cm$^{-3}$) and 227 K (with [HCl] = $3.3 \times 10^{10}$ molecule cm$^{-3}$). The concentration-

temperature pairs result in comparable, fractional equilibrium surface coverage of HCl on ice, with $\theta_{HCl} = 0.71 \pm 0.05$. The dataset indicates that the slope of $\gamma_{trap}$ versus the velocity of ice growth is largest at the lowest temperature. This result is expected, as the lifetime of an adsorbed [HCl] molecule at the ice surface will be longest at low temperatures where the rate



of thermally activated desorption is low. Longer surface residence times should increase the probability of trapping for any given ice growth velocity as the ratio of the rate of $H_2O$ uptake to the rate of HCl desorption increases. In Fig. 4a we plot $J_{HCl}$ instead of $\gamma_{trap}$ for the same dataset. Both the absolute flux and the efficiency of uptake of HCl show a similar dependence on temperature.

In order to facilitate the implementation of the experimental results into calculations of HCl interaction with growing ice surfaces under varying atmospheric conditions, we parametrised the data based on the dependence of $\gamma_{trap}$ on $V_{IG}$, $T$ and [HCl]. The basis of the parameterisation is the linear dependence of $\gamma_{trap}$ on $V_{IG}$ with values of ($\gamma_{trap}$ / $V_{IG}$) calculated for each dataset at constant temperature and [HCl]. The dependence of ($\gamma_{trap}$ / $V_{IG}$) on the HCl concentration (in the form of the calculated surface coverage, $\theta$ ) is displayed in Fig. 5 where data at the five different experimental temperatures are plotted.

The uptake coefficients tend to zero when the equilibrium surface coverage approaches unity and the fits to the data were forced through the origin. The rationale for this is the inherent assumption in the Langmuir isotherm that for $\theta_{HCl} = 1$ increasing the HCl concentration increases the collision frequency of HCl with the ice surface. The number of trapped HCl molecules is however limited by the ice growth velocity (Fig 3a) and the efficiency of trapping (number of trapped molecules divided by the collision frequency (equation 1) must tend to zero as [HCl] tends to infinity.

The temperature dependent slopes of ($\gamma_{trap}$ / $V_{IG}$)$(1-\theta_{HCl})^{-1}$ are plotted versus inverse-temperature in Arrhenius form in Fig.6. enabling the dataset to be parameterised with expression (5).

$$\gamma_{trap} = 220\exp(2025/T) \times V_{IG}(1-\theta_{HCl}) \qquad\qquad (5)$$

Equation (5) is valid for low ice growth rates where the uptake of HCl is not limited by gas-phase diffusion. The negative temperature dependence ($E$/R = 2025) indicates that the trapping of HCl in growing ice is associated with a net energy

change of ~ 17 kJ mol$^{-1}$. The fact that the fit line does not pass through all of the data-points plus the experimental uncertainty reflects the fact that the data were obtained on different ice surfaces. Although we attempted to make the ice as reproducible as possible, some scatter in $\gamma_{trap}$ was observed for different ice surface even under apparently identical conditions. This phenomenon has often been observed in studies of trace gas interaction with ice surfaces (Crowley et al., 2010).

A physical framework (trapping-model) describing the irreversible, nonreactive uptake of trace gases by growing ice particles has been developed by Kärcher and colleagues (Kärcher and Basko, 2004; Kärcher et al., 2009). The trapping-model enables the calculation of $\gamma_{trap}$, $J$ and $M$ from parameters $n^*$ (the maximum possible number density of molecules trapped in the ice) and $v_{esc}$ (an escape velocity which is connected to the surface desorption rate constant). Although the trapping-model was originally developed for nitric acid, it can in principle be used to describe the trapping of any molecule

in growing ice. Kärcher et al. consider the uptake of traces gases to growing ice surfaces to be a two-step process involving initial accommodation at the surface followed by trapping as new monolayers of water condense on the growing ice. The model distinguishes between different limiting cases which are characterised by different ratios of $v_{esc}$ to $V_{IG}$. If ($v_{esc}$ / $V_{IG}$) is





smaller than 0.1, i.e. the ice grows very rapidly or the desorption velocity is very small, $\gamma_{trap}$ approaches the mass accommodation coefficient ($\alpha$) of the respective trace gas such that every molecule that accommodates also gets trapped ("burial limit"). If ($v_{esc} / V_{IG}$) is larger than 10, most accommodated molecules will desorb from the ice surface rather than be trapped and $\gamma_{trap}$ is much smaller than $\alpha$ ("adsorption limit"). The experiments presented here may be considered to be at the

adsorption limit since the maximum value of the uptake coefficient was ~ 0.018, more than an order of magnitude lower than the mass accommodation coefficient of HCl on ice ($\alpha$ ~ 0.3, Crowley et al. (2010)). At the adsorption limit, $\gamma_{trap}$ and $J_{HCl}$ are predicted to scale with $V_{IG}$, which is in agreement with the experimental results of this study (Figure 3). Also, if HCl concentrations are sufficiently high, the flux of HCl to the surface is limited by $n^*$ so that $J_{HCl}$ is independent of [HCl] but $\gamma_{trap}$ decreases as [HCl] increases, also in accord with our observations.

To assess the trapping-model, we derive $n^*$ and $v_{esc}$ from our experimental datasets obtained at different temperatures and HCl concentrations. At the adsorption limit, the maximum number density of HCl in ice during trapping ($n^*$) can be calculated from the mole fraction $M_{HCl}$ (Kärcher and Basko, 2004; Kärcher et al., 2009) :

$$M_{HCl} = v_w \, \theta_{HCl} \, n^* \tag{6}$$

where $v_w$ is the volume occupied by an ice molecule in ice. Using mean values of $M_{HCl}$ from datasets obtained at constant ice

surface temperature and constant HCl concentration (variation of $V_{IG}$ only) we derive temperature dependent values of $n^*$ which are plotted in Arrhenius-form in Fig. 7 and described by:

$$n^* = 1.28 \times 10^{14} \exp(2090/T) \text{ molecule cm}^{-3} \tag{7}$$

which is represented by the solid fit-line through the data points. $n^*$ thus displays a significant, negative dependence on temperature. We also plot (right-hand y-axis) the solubility of HCl in ice (as a mole fraction for a HCl partial pressure

similar to those used in the present study) as measured by (Thibert and Dominé, 1997). Although their parameterization was derived in a different temperature range (265 – 238 K) and using single-crystalline ice, the temperature dependence is very similar. This highlights the fact that although $n^*$ cannot be equated to a solubility (which infers thermodynamic equilibrium), it is a related parameter.

Using equation (7), $n^*$ can be derived for any experimental temperature enabling $v_{esc}$ to be calculated from: (Kärcher and

Basko, 2004; Kärcher et al., 2009):

$$v_{esc} = \frac{\alpha \bar{c}}{4 K_{Lang} k_b T n^*} \tag{8}$$

where $k_B$ is the Boltzmann constant. Taking the present recommendations (IUPAC, 2019) for the Langmuir partition coefficient and accommodation coefficient (see above) we can derive $v_{esc} = 2.57 \times 10^{-3} \exp(-2090/T)$ m s$^{-1}$.

At 229 K equation (7) indicates a value of $n^*$ equal to $1.1 \times 10^{18}$ cm$^{-3}$, which is significantly larger ($\approx$ factor 10) than that

calculated by Kärcher et al. (2009) for HNO$_3$ trapping during ice-growth ($n^*$(HNO$_3$) = $1.5 \times 10^{17}$ molecule cm$^{-3}$). Our value of $v_{esc}$ at the same temperature is $2.8 \times 10^{-5}$ cm s$^{-1}$ which can be compared to the value of $1 \times 10^{-4}$ cm s$^{-1}$ calculated by



Kärcher et al. (2009). The values of $n^*$ and $v_{esc}$ derived in this work are based on an extensive laboratory dataset for HCl whereas those calculated by Kärcher et al. (2009) are derived from a very limited number of data for HNO$_3$ obtained under non-optimal conditions using a flow tube reactor in which the ice growth-rate was poorly characterized and probably non-homogeneous. A more detailed comparison of the values of $n^*$ and $v_{esc}$ between these studies is therefore not warranted.

However, in the following, we compare uptake coefficients of HCl ($\gamma_{trap}$) based on our expression (5) with those calculated using the trapping-model adapted for HCl by using values of $n^*$ and $v_{esc}$ derived from our data (equations (7) and (8)).

We calculate the trapping-model predictions of $\gamma_{trap}$ under conditions of planar flow (no diffusive limitation to uptake) at the adsorption limit and considering the limiting cases of high HCl partial pressures (surface saturation, $\gamma_{trap}^{sat}$) and low HCl-partial pressures (no surface saturation, $\gamma_{trap}^{unsat}$) whereby, according to (Kärcher et al., 2009):

$$\gamma_{trap}^{sat} = \frac{4v_{IG}n^*}{\bar{c}[HCl]} \tag{9}$$

$$\gamma_{trap}^{unsat} = \frac{\alpha v_{IG}}{v_{esc}} \tag{10}$$

The values of $\gamma_{trap}$ from equation (5) and the trapping-model are displayed in Fig. 8 which plots $\gamma_{trap}$ as a function of the HCl-partial pressure for two different temperatures (200 and 230 K) and at an ice growth velocity of 10 nm s$^{-1}$. The trapping-model accurately reproduces the empirical expression (5) once values of $n^*$ have been multiplied by a factor 1.3 (and thus

$v_{esc}$ decreased by the same factor, see equation (8)). The parameters used to calculate $n^*$ include the ice-growth velocity, the HCl concentration and the resultant fractional coverage of HCl on the surface so that a factor of 1.3 can be considered to be within experimental uncertainty. The fact that the trapping-model and the empirical parameterisation of our experimental data agree so well, reflects the fact that common values of the partition coefficient ($K_{Lang}$) have been used, and that both the trapping-model and our data analysis implicitly assume a common mechanism which involves surface accommodation and a

trapping process that competes with desorption. Our data serve to confirm the concept of a limiting, maximum concentration of HCl in growing ice that is related to but significantly larger than the solubility of HCl in single ice crystals and thus provide validation of the underlying physical processes in the trapping model (Kärcher and Basko, 2004; Kärcher et al., 2009).

### 3.1.1 Previous experimental data on the uptake of HCl to growing ice surfaces

The uptake of HCl to equilibrium ice surfaces has been extensively studied and the results summarised by the IUPAC panel (Crowley et al., 2010; IUPAC, 2019). A limited number of laboratory studies have explored the uptake of HCl trace gases during ice growth, though generally not under conditions that enable detailed comparison with the present dataset or analysis with the trapping-model.

Santachiara et al. (1995) studied the trapping of HCl during ice-growth at 260 K, whereby ice crystals were grown on a

metal surface via the Wegener-Bergeron process by transfer of water from super-cooled water droplets which contained





hydrochloric acid. Analysis of the HCl-content of the ice particles indicated up to 100% transfer of HCl from the evaporating aqueous droplets to the growing ice crystals, with HCl mole fractions in ice ($M_{HCl}^{ice}$) of $8.5 \times 10^{-5}$ to $4 \times 10^{-3}$. At 260 K, such concentrations are in an area of the ice/HCl phase diagram where the solid HCl-ice solution coexists with an aqueous phase (Thibert and Dominé, 1997). While these experiments indicate that HCl is incorporated efficiently into growing ice,

quantitative comparison with our data is precluded by a lack of information pertaining to the HCl concentration in the gas-phase, the temperature of the ice or the rate of ice growth.

Diehl et al. (1995) grew ice crystals directly from the gas phase onto a cooled filament. The ice was grown for 9-18 hrs at 258 K and at 15% super-saturation with HCl concentrations in the gas phase between $2.8 \times 10^{11}$ and $9.9 \times 10^{12}$ molecule cm$^{-3}$. Analysis of the ice-phase indicate that, up to a value of up to $0.25 \times 10^{-6}$, $M_{HCl}^{ice}$ was linearly proportional to the

concentration of HCl, increasing further to a maximum value of $1 \times 10^{-6}$ as [HCl] was increased further. The fact that the rate of ice-growth is unknown in the experiments of Diehl et al. (1995) and the use of high pressures and diffusion limited uptake preclude a more quantitative comparison.

Dominé and Rauzy (2004) studied HCl trapping in ice under conditions similar to those employed by Diehl et al. (1995) and were thus subject to the same limitations regarding diffusion, precluding a kinetic analysis. In their work, the lowest growth

velocity was about 1000 nm s$^{-1}$ and thus 25 times higher than the highest growth velocity in our experiments. A combination of fast ice growth and high pressure in their experiments will have resulted in substantial depletion of gas-phase HCl close to the ice surface, effectively limiting the transfer of HCl to the surface and thus reducing $M_{HCl}^{ice}$. In accordance, the authors report a decrease in $M_{HCl}^{ice}$ with increasing growth velocity.

Huthwelker (1999) investigated the uptake of HCl on growing ice using a Knudsen reactor at temperatures of 190 and 203

K. The gas phase concentrations of HCl were between of $1 \times 10^{8}$ to $1 \times 10^{11}$ molecule cm$^{-3}$ while ice growth velocities were between 0.2 and 100 nm s$^{-1}$. As in the present study, the mass spectrometric detection of HCl and H$_2$O enabled the real time measurement of $\gamma_{trap}$ and $M_{HCl}$. In qualitative agreement with this study, Huthwelker (1999) measured comparable values of $J_{HCl}$ under similar conditions and observed that $\gamma_{trap}$ and $M_{HCl}^{ice}$ increase with $V_{IG}$. However, while we find that $J_{HCl}$ is proportional to $V_{IG}$, Huthwelker (1999) found $J_{HCl} \propto V_{IG}^{1.5}$ which makes $M_{HCl}^{ice}$ approximately proportional to $V_{IG}^{0.5}$. Another

major difference between the two studies is the temperature dependence of $\gamma_{trap}$. While Huthwelker (1999) found no statistically significant variation in $\gamma_{trap}$ between 190 and 203 K our data suggest that $\gamma_{trap}$ doubles in this temperature range. Though the reason for these discrepancies is obscure, we note that a major difference between the two experiments lies in the mode of preparation of the ice film. The vapour deposited ice films of Huthwelker (1999), are rough / porous, and, under equilibrium, the dominant fraction of HCl uptake was attributed to bulk-diffusion rather than surface adsorption. In contrast

the ice used in this study, prepared from the liquid phase, is thought to be smooth on a microscopic level and exhibited hardly any diffusive uptake. It remains unclear if and how the different ice formation modes and ice morphologies modify the trapping process.



**3.2 Uptake of oxidized organics: ethanol, 1-butanol, formic acid and tri-fluoro acetic acid**

Analogous to the experiments with HCl described above, we examined the uptake of ethanol, 1-butanol, formic acid and tri-fluoro acetic acid to growing ice surfaces. Despite the use of high values of $V_{IG}$, no sign of trapping (i.e. a net reduction in the flux of molecules to the mass-spectrometer during ice-growth) was observed for ethanol, 1-butanol or formic acid. Tri-fluoro acetic acid represents an intermediate case, exhibiting continuous uptake during ice growth (213 and 223 K) with $\gamma_{trap}$ positively correlated to $V_{IG}$ and the largest value of $\gamma_{trap}$ obtained at the lowest temperature. The derivation of $\gamma_{trap}$ for tri-fluoro acetic acid is however associated with large uncertainty as the change in concentration is small and small baseline drifts and / or changes in the mass spectrometer sensitivity increase the uncertainty to a factor of ~2. For this reason, we restrict our discussion of the tri-fluoro acetic acid dataset (intermediate $\gamma_{trap}$) to qualitative comparison with HCl (large $\gamma_{trap}$) and the other oxidised organic trace gases (low $\gamma_{trap}$).

The results for the organic traces gases are summarised along with those for HCl in Table 1, which also lists $V_{IG}$, the range of fractional surface coverages and the approximate, average lifetime ($\tau_{des}$) with respect to desorption for each trace gas at the respective experimental temperature.

$$\tau_{des} = \frac{4 K_{Lang} N_{max}}{\alpha \bar{c}} \tag{11}$$

Values of $K_{Lang}$ and $\alpha$ (0.3) were taken from an evaluation (Crowley et al., 2010; IUPAC, 2019). Despite the deposition of hundreds to thousands of monolayers of water on the ice surface during the desorption lifetime, no trapping of ethanol, 1-butanol or formic acid could be observed with $\gamma_{trap} < 6 \times 10^{-5}$, i.e. more than two orders of magnitude lower than for HCl for similar ice growth rates and surface residence times. In qualitative agreement with our results, Abbatt et al. (2008) were unable to observed trapping of ethanol during ice-growth. Huffman and Snider (2004) studied the uptake of ethanol and 1-butanol to ice at 249 to 269 K which was growing with velocities of 24 to 110 nm s$^{-1}$. The volume uptake coefficients reported (amount of gas taken up per partial pressure and ice mass), displayed a positive T-dependence and were independent of the ice growth velocity. The positive T-dependence of the uptake observed by Huffman and Snider (2004) points to an uptake mechanism that is different from the one we observe for HCl, with dissolution of the organic trace gases in a quasi-liquid layer on the ice surface or in grain boundaries and triple junctions playing a potentially dominant role as the ice temperature approached the melting point. Taking the trace-gas partial pressures used by Huffman and Snider (2004), and extrapolating their T-dependent uptake coefficient to 228 K results in mole fractions of $3 \times 10^{-10}$ for ethanol and $3 \times 10^{-9}$ for 1-butanol in the ice. The differences in gas-phase concentrations leading to such low molar fractions would have been too small to be quantifiable with the methods used in this study.

The differences in $\gamma_{trap}$ between these trace-gases provides an important clue to which factors may drive trapping of a physi-adsorbed trace gas on a growing ice surface at temperatures far below the melting point. Studies of the equilibrium partitioning of ethanol, butanol and formic acid (and other oxidised organics) between ice and gas-phase indicate that they are adsorbed in molecular form via hydrogen bonds (Compoint et al., 2002; Jedlovszky et al., 2006; Pártay et al., 2007;





Jedlovszky et al., 2008) and that the uptake is fully reversible (Sokolov and Abbatt, 2002; Peybernès et al., 2004; Kerbrat et al., 2007; von Hessberg et al., 2008). In contrast, the uptake of HCl to ice is known to be partially irreversible and there is overwhelming experimental and theoretical evidence that HCl dissociates upon adsorption (Gertner and Hynes, 1998; Svanberg et al., 2000; Toubin et al., 2002; Park and Kang, 2005; Ayotte et al., 2011; Parent et al., 2011). For tri-fluoro acetic

acid the adsorption is partially dissociative (Symington et al., 2010) and displays similar features to HCl, including partially non-reversible adsorption on pristine ice surfaces and partitioning coefficients that are independent of temperature (Symington et al., 2010; Zimmermann et al., 2016). Table 1 lists the acid dissociation constants ($pK_a$) for several of the trace-gases investigated in thus study, highlighting the significant difference between the acidity of HCl (pKa = -7) per-fluoro acetic acid (pKa = 0.52) and formic acid (pKa = 3.75).

We thus conclude that, following surface accomodation, the dissociative formation of ions is prerequisite for efficient trapping at the temperatures and ice-growth rates of our study. This implies that ice lattice formation around solvated hydronium cations and acid anions is thermodynamically favoured, whereas physi-adsorbed trace-gases impede growth of ice at the adsorption site. While such dynamical effects are not included in the trapping model (see above) they are empirically incorporated in the parameters $n^*$ and $v_{esc}$.

### 3.3 HCl trapping on atmospheric ice particles

Our laboratory dataset and the parameterisation of the trapping of HCl in growing ice surface enables us to calculate the particle-phase HCl-concentration associated with an ice cloud at a given ice-particle growth rate and different ice-particle diameters. This can be compared to the particle-phase concentration of HCl resulting from partitioning of HCl to the surface of ice crystals and to the solubility of HCl in ice under equilibrium conditions when the ice-growth velocity is zero. In each

case we calculate the mole fraction of HCl associated with the ice-phase for a given gas-phase concentration and temperature.

As HCl-uptake to the ice is a non-equilibrium effect described by an effective uptake coefficient we need to consider the limiting effect of gas-phase diffusion to the surface. To do this we modify the values of $\gamma_{trap}$ obtained in equation (5) using equation (12):

$$\gamma_{trap}^{dif} = \left( \frac{1}{\gamma_{trap}} + \frac{0.75 + 0.283 K_n}{K_n(K_n + 1)} \right)^{-1} \qquad (12)$$

where $K_n$ is the Knudsen number and is equal to $3D_{HCl}/\bar{c}r$. $D_{HCl}$ is the temperature and pressure dependent diffusion coefficient of HCl in air and $r$ is the particle radius. $D_{HCl}$ was calculated using the Fuller equation (Fuller et al., 1966) and literature values for the diffusion volumes of HCl and air (Tang et al., 2014).

Figure 9 compares the contributions of surface adsorption ($V_{IG} = 0$, black lines) and trapping of HCl ($V_{IG} = 10$ nm s$^{-1}$, blue

lines) to the mole-fraction of particulate HCl for prescribed conditions of HCl-partial pressure ($1 \times 10^{-7}$ Torr) and total pressure (100 Torr) at temperatures that may be found in the upper troposphere and at the tropopause. The HCl mole-





fractions associated with particles of radius 2, 6 and 18 microns resulting from surface adsorption were calculated using the Langmuir isotherm with $K_{Lang} = 9.6 \times 10^{-11}$ cm$^3$ molec$^{-1}$, independent of temperature (IUPAC, 2019). The contribution of the initial "pure" ice particle to the mole fraction of water was neglected during these calculations as the initial particle volume is small compared to that obtained after several minutes of growth in the presence of HCl. We also plot the mole fraction of

HCl resulting from bulk solubility in ice (red curve), according to Thibert and Dominé (1997).

Figure 9 illustrates several key differences in the kinetically controlled trapping process compared to the equilibria associated with surface adsorption or solubility. The HCl-mole-fraction associated with trapping displays a strong dependence on temperature but only a weak dependence on particle diameter. In contrast, Langmuir adsorption, a surface-only effect, displays no dependence on temperature but a strong negative dependence on diameter, the latter reflecting the

fact that the volume of a growing particles increases proportional to $r^3$, whereas the surface area increases proportional to $r^2$. Trapping can therefore greatly enhance the overall uptake of HCl for particles of radius larger than 6 μm and thus strongly influence the gas- to particle partitioning during particle growth. The surface contribution to the HCl mole-fraction is relatively more important only for smaller particles and for periods when the ice growth rate is low.

The solubility limit for HCl imbedded in an ice single crystal (Thibert and Dominé, 1997) is shown in red for reference. The

values for the mole fraction of HCl in ice obtained during our experiments clearly exceed the thermodynamic limit. This likely reflects the fact that the time-scale required to acquire equilibrium (i.e. diffusion of trapped HCl back to the gas-phase) are much longer than the time-scales of our experiments. The diffusion coefficient of HCl in ice is $< 10^{-12}$ cm$^2$ s$^{-1}$ (see Thibert and Dominé (1997) and references therein) implying timescales for relaxation to equilibrium of $>10^5$ s for particles of $10 \times 10^{-6}$ m radius, which are longer than the timescales associated with atmospheric temperature fluctuations which

control ice growth (and evaporation) rates.

An important difference between the uptake of HCl via equilibrium surface partitioning and trapping is the potential, fractional depletion of gas-phase HCl. In the case of surface adsorption, the system will relax to the equilibrium state in which the rates of adsorption and desorption of HCl to/from the ice surface are equal; the fraction of HCl remaining in the gas-phase depending on the initial concentration and the particle number-density. In the case of non-equilibrium trapping,

the complete transfer of the HCl to the particle-phase during ice-growth can occur. Consider a cirrus cloud (230 K) with an initial ice-particle surface area density ($A$) of $2 \times 10^{-4}$ cm$^2$ cm$^{-3}$ (see e.g. (Popp et al., 2004)) and ice particles growing at 10 nm s$^{-1}$. The uptake coefficient for HCl ($1 \times 10^{-8}$ Torr) is $\gamma_{trap} \sim 4 \times 10^{-3}$, which converts (via equation 13) to a lifetime for gas-phase HCl ($\tau$) of only a few minutes.

$$\tau_{HCl} = \frac{4}{\gamma_{trap} \bar{c} A} \tag{13}$$

The efficient removal of HCl from the gas-phase in ice clouds via the trapping mechanism can have important repercussions for heterogeneous reactions taking place on the ice surface. While surface adsorbed HCl is available for reactions with e.g. ClONO$_2$, HOCl and N$_2$O$_5$, which liberate photo-chemically active forms of chlorine (Crowley et al., 2010), HCl that is



trapped in the bulk of the ice will likely not participate in these reactions until the ice starts to evaporate. This may influence the spatial and temporal activation of chlorine and may likewise influence the development of the polar ozone hole.

## 4 Conclusions

Our experiments have clearly demonstrated that HCl is incorporated (trapped) efficiently in growing ice surfaces. The coefficient ($\gamma_{trap}$) describing the uptake of HCl to an ice surface during growth was proportional to the velocity of ice-growth and was largest at low temperatures when the desorption lifetime of HCl accommodated at the ice surface is longest. $\gamma_{trap}$ was found to be reduced at high HCl partial-pressures and our experimental results are entirely consistent with the framework for trapping developed by Kärcher and colleagues (Kärcher and Basko, 2004; Kärcher et al., 2009).

No experimental evidence for the trapping of several oxidised organics that bind to the ice-surface via H-bonds could be found, whereas tri-fluoro acetic acid, which partially dissociates on the ice surface, was an intermediate case. We infer that the efficient trapping of HCl is related to its dissociation and solvation in the growing ice-lattice, the mole-fraction of HCl exceeding the thermodynamic solubility limit.

Our parameterisation of the experimental data enabled us to compare the relative importance of HCl-trapping during ice-growth with equilibrium partitioning due to surface uptake only, and will facilitate the incorporation of HCl-trapping in atmospheric models. Trapping of HCl in ice particles during their growth can represent a highly efficient route to removal of HCl from the gas-phase in ice clouds, which, in contrast to surface-only-uptake, does not make HCl available for heterogeneous, surface reactions.

## Data availability

The underlying data (Figures 3-7) are tabulated in the supplementary information.

## Author contributions

Matthias Kippenberger carried out the experiments and analysed the data. John Crowley, Matthias Kippenberger and Jos Lelieveld wrote the manuscript. Gerhard Schuster designed and built the reactor for investigating trace-gas uptake to growing ice surfaces and helped with data interpretation.

## Competing interests

The authors declare that they have no conflict of interest.



## Acknowledgements

This study was carried out in partial fulfilment of a doctoral dissertation (M. Kippenberger) at the University of Mainz, was partially funded by the International Max-Planck Research School and contributes to the MaxWater initiative of the Max Planck Society. We thank Chemours for provision of the FEP sample (FEPD 121) used to coat the reactor.

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





## Table 1: Summary of results.

| compound | $T$ (K) | $\tau_{des}$ (s) | $pKa$ | $V_{IG}$ (m s$^{-1}$) | $\theta_{HCl}$ | $\gamma_{trap}$ |
|---|---|---|---|---|---|---|
| ethanol | 214 | 0.4 | 18[a] | 2.0x10$^{-8}$ to 1.8x10$^{-7}$ | 0.2 to 0.8 | < 6.0x10$^{-5}$ |
|  | 219 | 0.2 |  |  |  |  |
| 1-butanol | 209 | 20.3 | 17[a] | 2.0x10$^{-8}$ to 2.3x10$^{-7}$ |  |  |
|  | 223 | 1.3 |  |  |  |  |
|  | 228 | 0.5 |  |  |  |  |
| formic acid | 213 | 3.2 | 3.75[b] | 5.0x10$^{-8}$ to 1.3x10$^{-7}$ | 0.4 to 0.7 | < 7.0x10$^{-5}$ |
| tri-fluoro acetic acid | 213 | 14.5 | 0.52[b] | 4.0x10$^{-9}$ to 4.5x10$^{-8}$ | 0.5 to 0.9 | 5.2x10$^{-4}$ to 3.4x10$^{-3}$ |
|  | 223 | 14.2 |  | 4.5x10$^{-9}$ to 4.3x10$^{-8}$ | 0.5 | 2.0x10$^{-6}$ to 4.5x10$^{-4}$ |
| HCl | 213 | 55.8 | -7[c] | see text for details |  |  |

[a] $pK_a$ value from Becker (2001). [b] $pK_a$ value from Haynes and Lide (2001). [c] $pK_a$ value from Riedel (2002). The HCl desorption lifetime, $\tau_{des}$, was calculated from the recommended IUPAC values of $K_{Lang}$ and $\alpha$; for tri-fluoro acetic acid, the data of Symington et al. (2010) were used.





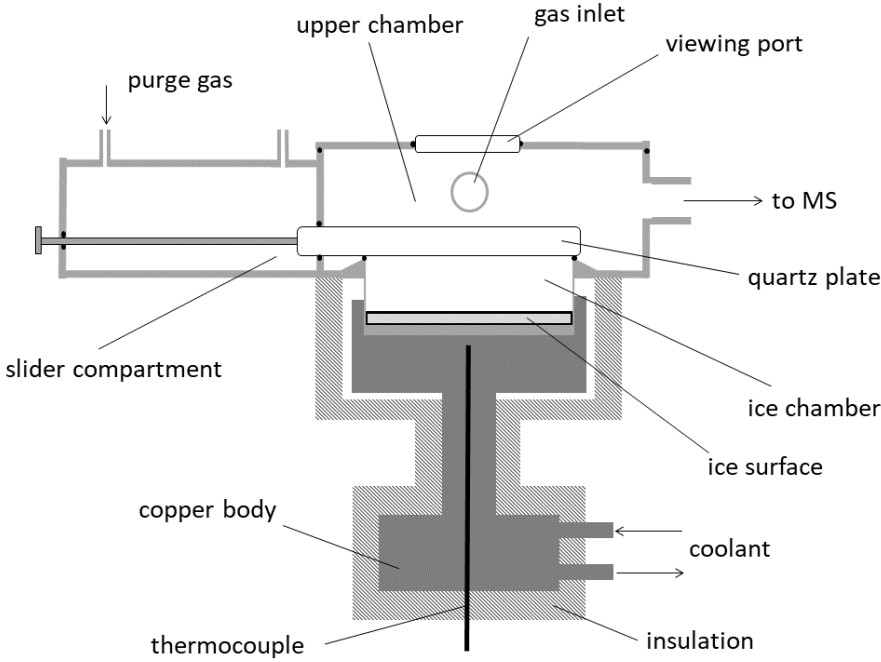

**Figure 1:** Schematic representation of the reactor used during the trapping experiments. An electrical heating element (not shown) is wrapped around slider compartment and upper chamber to keep them at room temperature.

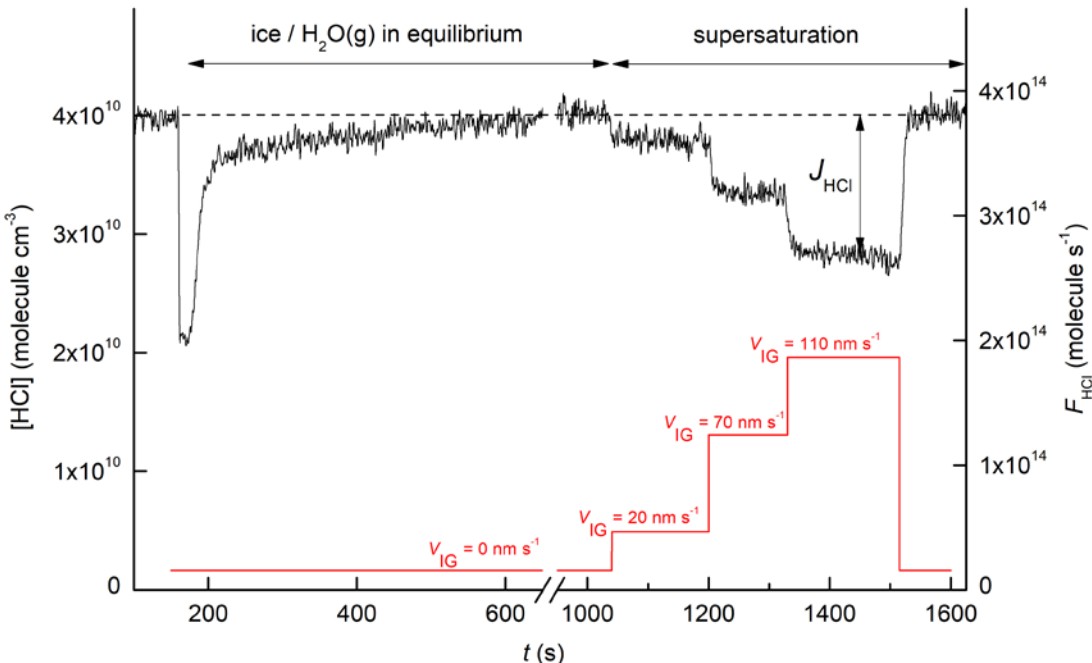

**Figure 2**: Black line: Time series of the HCl concentration (left y-axis) or flux (right y-axis) during an experiment at 213 K. Prior to $t = 1040$ s, the ice was in equilibrium with the gas-phase $H_2O$ concentration. After t = 1040 s, the ice was supersaturated with respect to the gas-phase and the red line indicates the ice growth velocity, $V_{IG}$ (based on the $H_2O$ signal at $m/z$ 20) during the experiment.

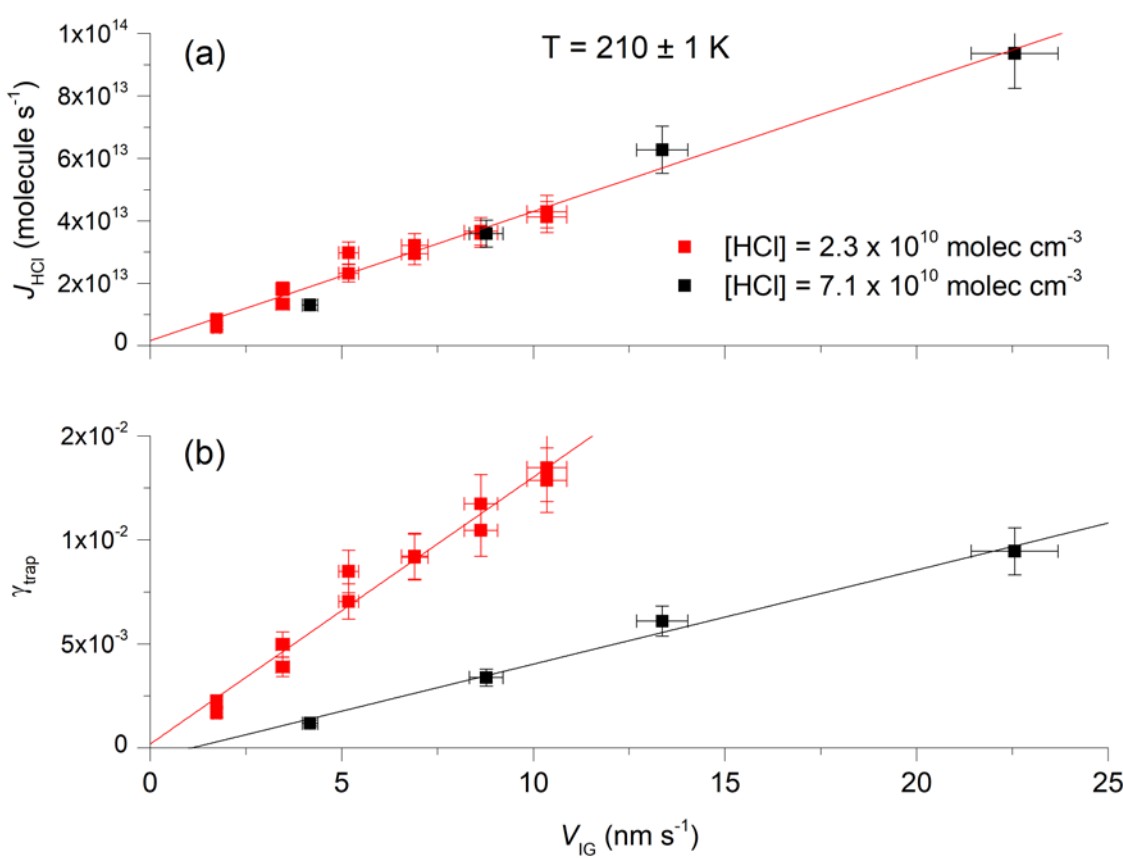

**Figure 3:** The trapping coefficient ($\gamma_{trap}$) as a function of ice growth velocity at 210 K using two different concentrations of HCl. The solid, straight lines are least-squares fits to the data. The error bars represent total uncertainty in $\gamma_{trap}$ and the ice growth velocity.





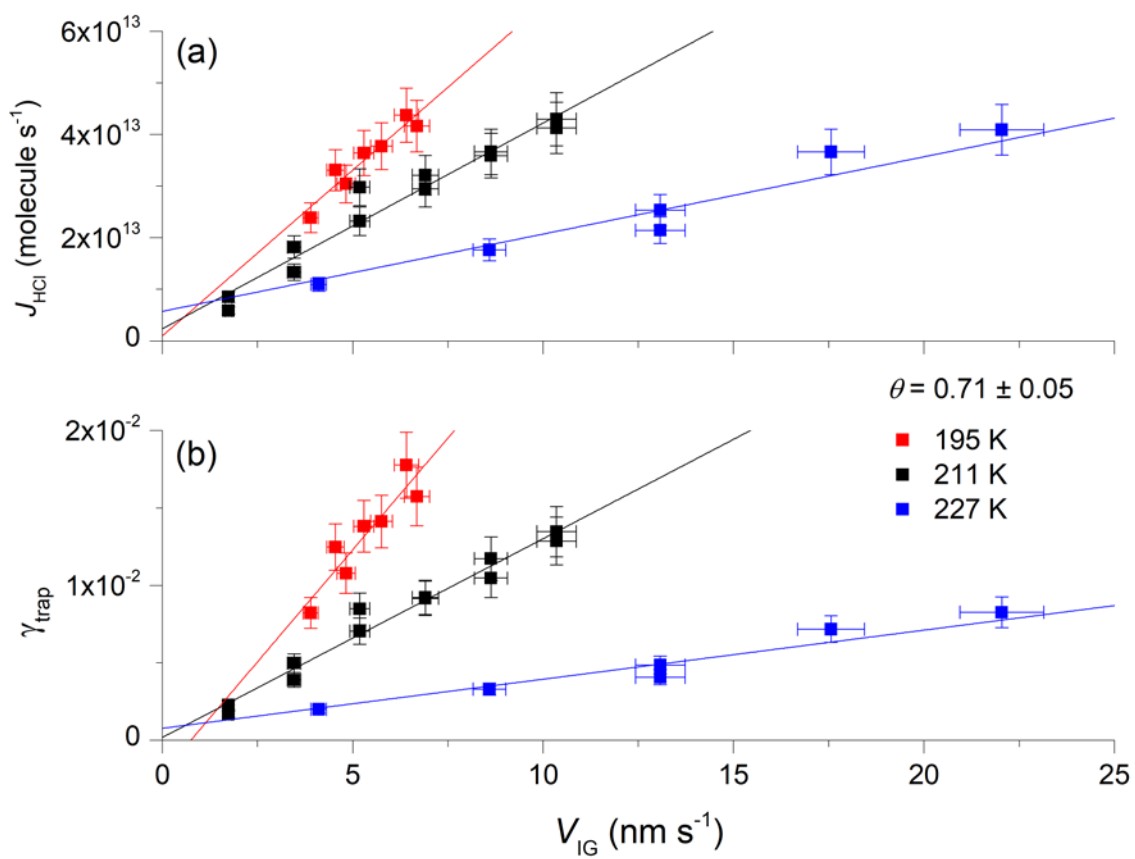

**Figure 4**: Dependence of the trapping coefficient, $\gamma_{trap}$, on the ice growth velocity ($V_{IG}$) at three different temperatures and a comparable equilibrium surface coverage ($\theta$) of HCl. The solid, straight lines are least-squares fits to the data. The error bars represent total uncertainty in $\gamma_{trap}$ and the ice growth velocity.



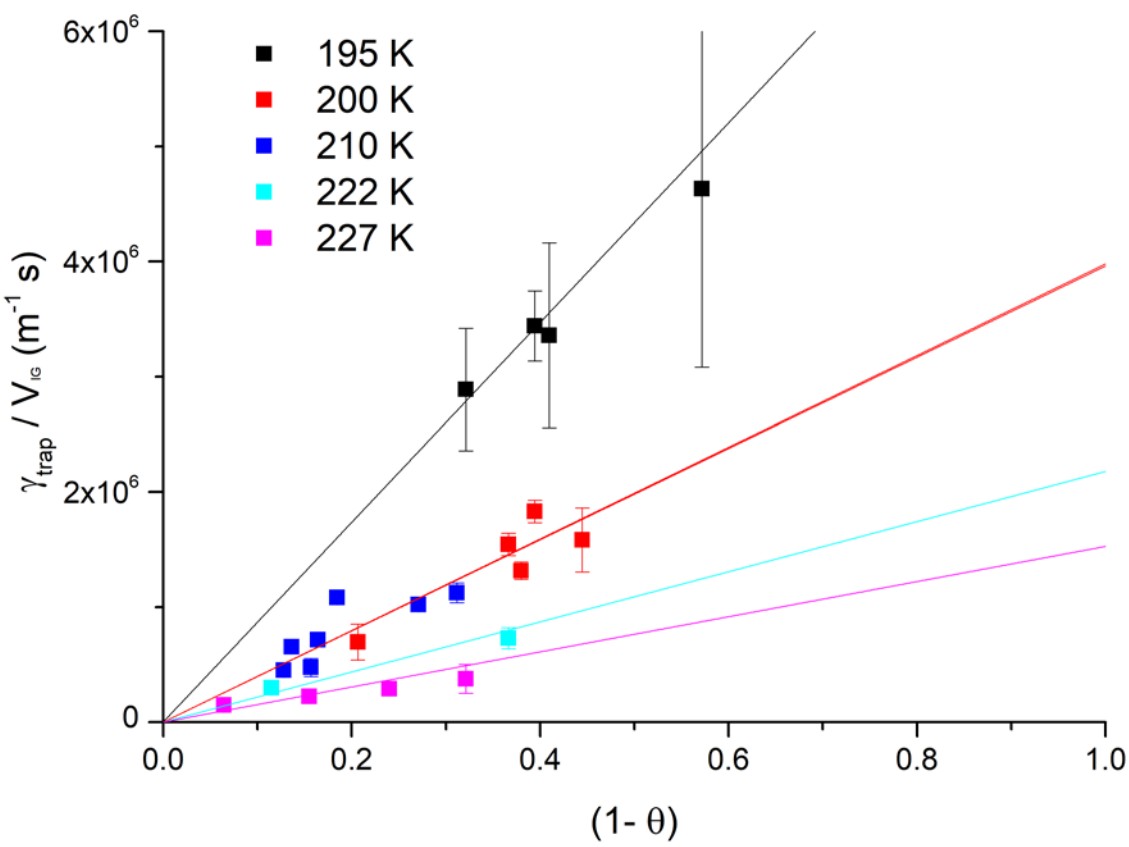

**Figure 5**: Plot of $\gamma_{trap} / V_{IG}$ (as derived from data in Figure 4) vs. (1-$\theta$) at different temperatures. The solid lines are proportional fits to the data at each temperature, the error-bars represent total uncertainty.

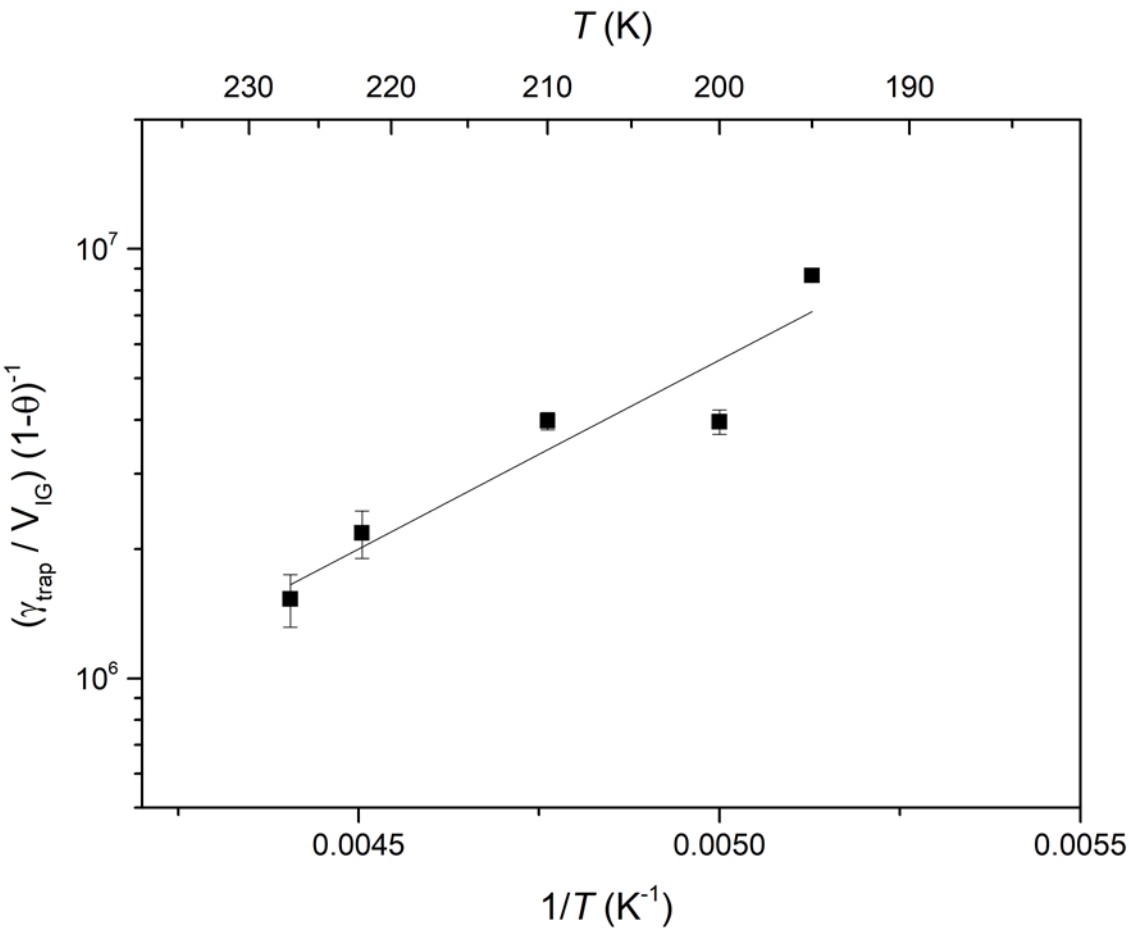

**Figure 6**: Arrhenius plot of $(\gamma_{trap} / V_{IG})(1-\theta)^{-1}$ derived from data in Figure 5 versus inverse temperature. The solid line is a least squares fit to the data.



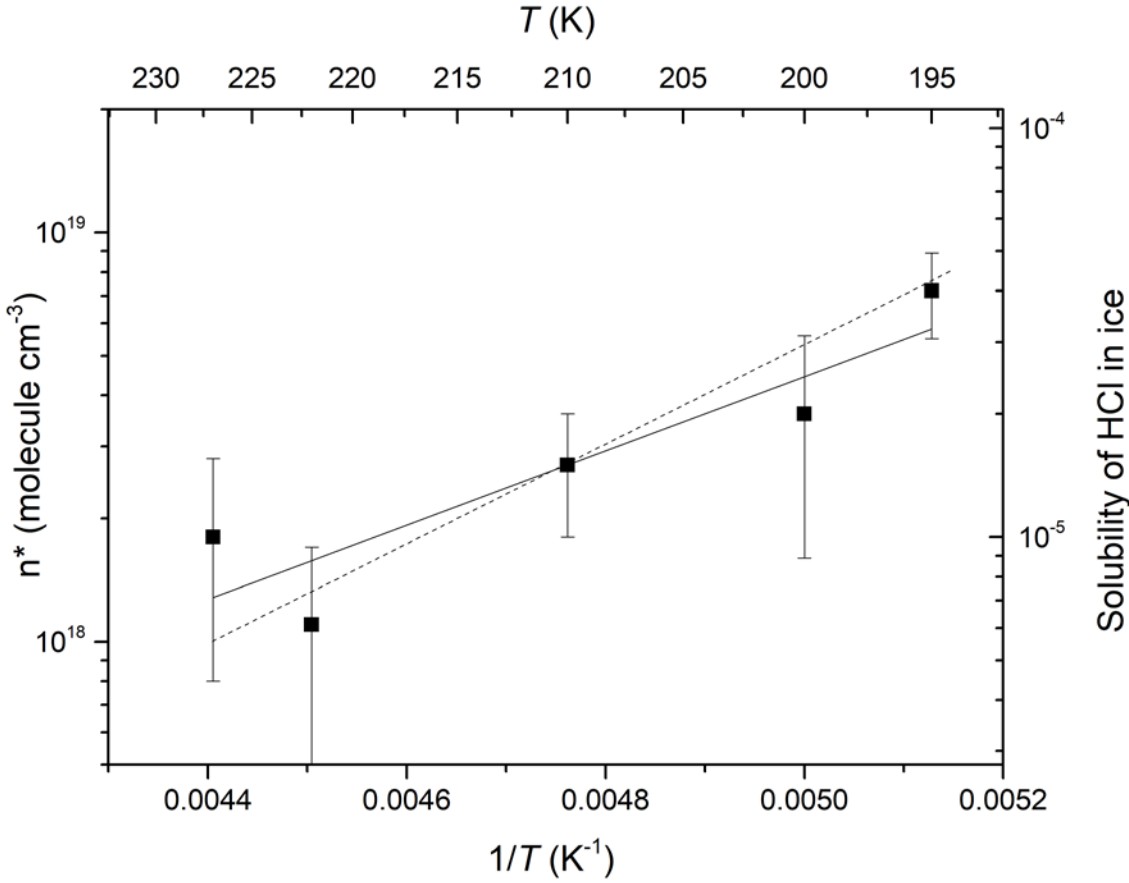

**Figure 7:** Arrhenius plot of $n*$ vs. inverse temperature. The error bars represent the $1\sigma$ standard deviation of the mean values derived from $M_{HCl}$. The solid line is a fit to the data according to expression (7). The dotted line (right-hand y-axis) represents the solubility of HCl in ice (as a unit-less mole fraction for a HCl partial pressure similar to those used in the present study) as measured by (Thibert and Dominé, 1997). The left- and right-y axes have the same relative span to enable comparison of slopes.

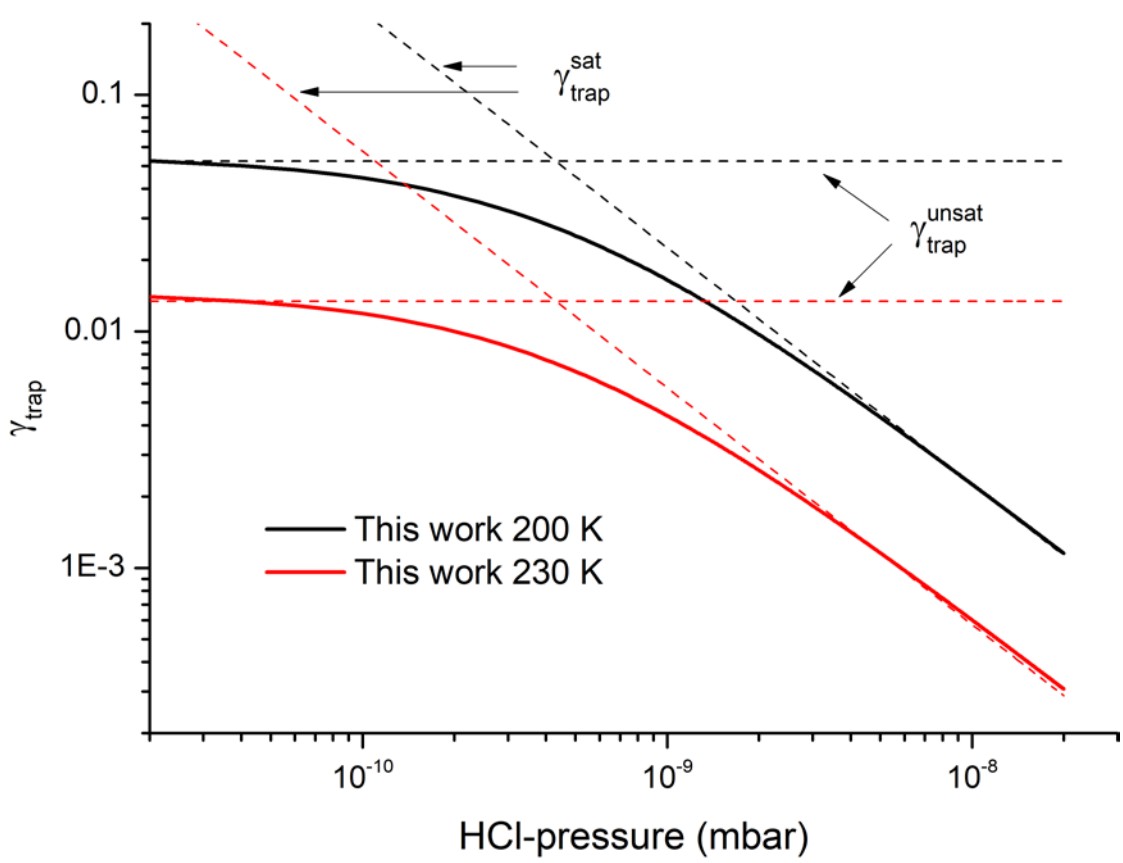

**Figure 8:** Comparison of $\gamma_{trap}$ from the present dataset, parameterised according to equation (5) with the results of the modified trapping-model from Kärcher et al (2009) for the limiting cases of saturated and unsaturated uptake. The values of $n^*$ derived from our data and used to calculate $v_{esc}$ and $\gamma_{trap}$ have been multiplied by a factor of 1.3 to improve agreement (see text).



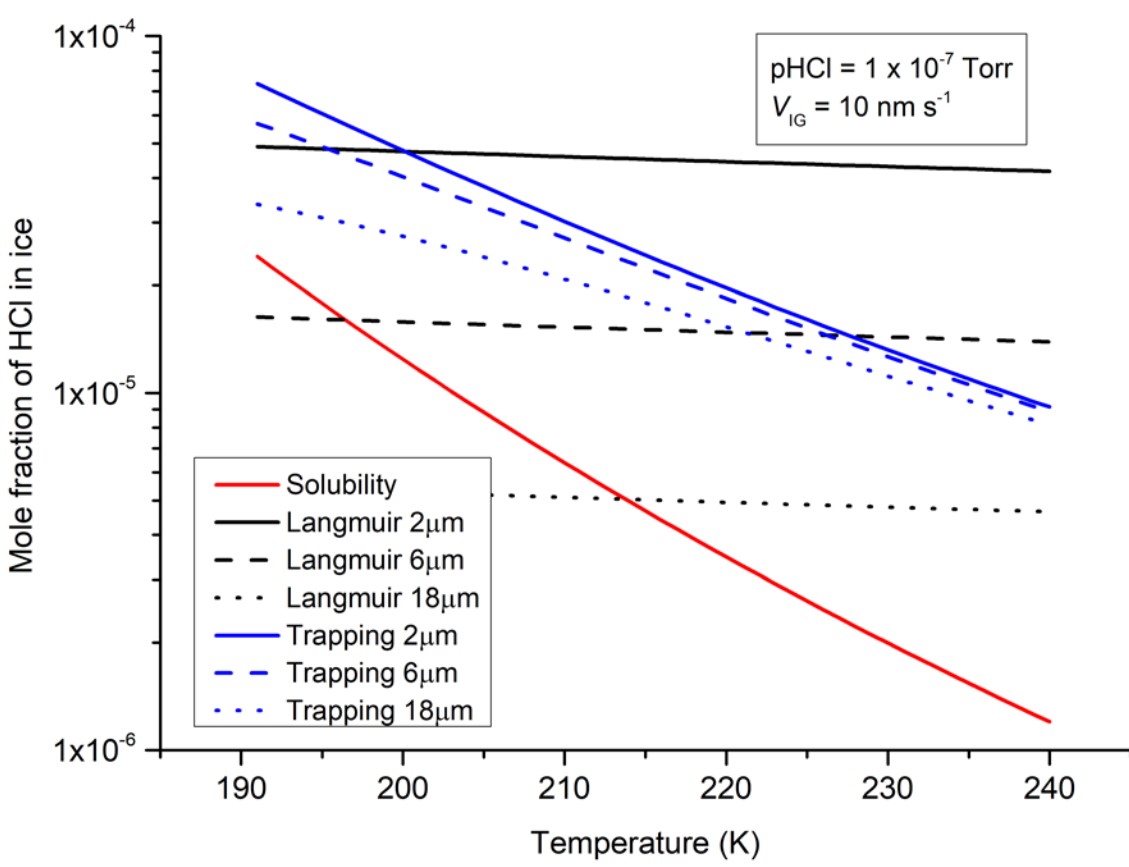

**Figure 9:** Molar fraction of HCl (for a gas-phase partial pressure = 1 x $10^{-7}$ Torr) associated with ice particles (2, 6 and 18 μm radius) considering either surface adsorption (Langmuir, zero ice growth) or trapping (ice growth velocity = 10 nm s$^{-1}$).