# Peer review of "Trapping of HCl and oxidized, organic trace-gases in growing ice at temperatures relevant for cirrus clouds."

_Atmospheric Chemistry and Physics, 2019_

## Referee Comment (RC1) · Anonymous Referee #1 · 17 Jul 2019

The authors present measurements of the uptake flux of HCl vapor (and other gases) to growing ice surfaces. The experiments were conducted under cirrus cloud conditions of temperature and ice particle growth rate, and the findings are that HCl may become buried by growing particles potentially immobilizing it from heterogeneous activation reactions. The uptake appears to be due to the need to hydrate dissociated ions, given that non-dissociating gases did not exhibit enhanced uptake under growing conditions.

Experiments are conducted in a new cylindrical dual-chamber through which a mixture of $H_2O$ and HCl in He is flowing in a warm section. At the bottom of the chamber is ice that has formed by freezing liquid water. A glass plate separating the warm and icefilled sections of the chamber is removed, allowing for exposure of the gases to the ice surface. The gas-phase water partial pressure is sufficient that water condenses, forming ice at a known rate based on the supersaturation. A mass spectrometer monitors the gas-phase composition. The data appear to be of very high quality, for example of uptake flux as a function of growth rate (Fig 3 and 4). The execution of the experiments and the interpretation of the data, done in the context of a trapping model developed by B. Karcher, are clearly described. There is good qualitative agreement between the model and the measurements. This is high quality work. I recommend publication with very few comments:

It is not clear to me how the gas-phase diffusion limitation for HCl uptake to the ice is accounted for. In particular, in a Knudsen Cell, which resembles the current cell in some respects, the diffusion constraint is minor because the pressure is kept quite low (about 10ˆ-3 mbar). But in this experiment the pressure is quite high, about a mbar. Shouldn't there be a large mass transfer limitation for getting the HCl to the ice surface under such conditions, when the intrinsic HCl uptake coefficient/mass accommodation coefficient to the ice is large? The authors say that such effects kick in at very high gamma values (top of page 6) and that they restrict their expts to gammas smaller than 0.018, but I would have guessed the effect is still important at much lower gammas. How was this value of 0.018 chosen? A full discussion of this mass transfer effect is warranted given how central it is to the measurements, and that this is a new experimental approach. On this topic, why is the chamber designed to operate at 1 mbar total pressure? Wouldn't it have been better to run at lower pressure to lessen the diffusion limitation?

I will be up-front and say that I have not worked through the Karcher trapping model in depth. But my impression is that it is based on a Langmuir adsorption isotherm, as a starting point. If so, to what degree does dissolution in the ice crystals themselves play a role in the uptake, both in the model and in the experimental data? Are thermodynamically stable solutions forming (see work by Domine), or are metastable

mixtures forming? If metastable, how long would it take for the enhanced "dissolved" HCl to diffuse out, under non-growth conditions? Also, is there uptake along the grain boundaries that may affect the (model or experimental) results?

How pure is the water that is condensing? Is there any potential for trace ammonia, for example, to be interacting with the sorbed HCl, so enhancing the uptake?

Not much experimental work has been done on uptake of gases to growing ice in the last few years, and the quality of the earlier studies is not nearly as high as in this work. Nevertheless, I suggest that a brief summary of past studies, upon which this work is somewhat based, could be presented in the Introduction.
* * *

---

## Author Comment (AC1) · 9 Aug 2019

**Referee 1**

In the following, the referee's comments are reproduced (black) along with our replies (blue) and changes made to the text (red) in the revised manuscript.

**General Comments:**

The authors present measurements of the uptake flux of HCl vapor (and other gases) to growing ice surfaces. The experiments were conducted under cirrus cloud conditions of temperature and ice particle growth rate, and the findings are that HCl may become buried by growing particles potentially immobilizing it from heterogeneous activation reactions. The uptake appears to be due to the need to hydrate dissociated ions, given that non-dissociating gases did not exhibit enhanced uptake under growing conditions.

Experiments are conducted in a new cylindrical dual-chamber through which a mixture of H2O and HCl in He is flowing in a warm section. At the bottom of the chamber is ice that has formed by freezing liquid water. A glass plate separating the warm and ice-filled sections of the chamber is removed, allowing for exposure of the gases to the ice surface. The gas-phase water partial pressure is sufficient that water condenses, forming ice at a known rate based on the supersaturation. A mass spectrometer monitors the gas-phase composition. The data appear to be of very high quality, for example of uptake flux as a function of growth rate (Fig 3 and 4). The execution of the experiments and the interpretation of the data, done in the context of a trapping model developed by B. Karcher, are clearly described. There is good qualitative agreement between the model and the measurements. This is high quality work. I recommend publication with very few comments:

We thank the referee for this very positive assessment of our manuscript.
* * *
**Specific Comments:**

It is not clear to me how the gas-phase diffusion limitation for HCl uptake to the ice is accounted for. In particular, in a Knudsen Cell, which resembles the current cell in some respects, the diffusion constraint is minor because the pressure is kept quite low (about 10^-3 mbar). But in this experiment the pressure is quite high, about a mbar. Shouldn't there be a large mass transfer limitation for getting the HCl to the ice surface under such conditions, when the intrinsic HCl uptake coefficient/mass accommodation coefficient to the ice is large? The authors say that such effects kick in at very high gamma values (top of page 6) and that they restrict their expts to gammas smaller than 0.018, but I would have guessed the effect is still important at much lower gammas. How was this value of 0.018 chosen? A full discussion of this mass transfer effect is warranted given how central it is to the measurements, and that this is a new experimental approach. On this topic, why is the chamber designed to operate at 1 mbar total pressure? Wouldn't it have been better to run at lower pressure to lessen the diffusion limitation?

There were several reasons for operating at a pressure of 1 Torr (1.33 mbar). This first was simply that the differential pumping system of the mass spectrometer was set up for these pressures (as used in our flow-tube studies e.g. the Zimmermann et al paper on HCl uptake which we cite) and the efficiency of detection was optimised for these conditions. Second, the lower limit to the pressure in the reactor is given by the vapour pressure of ice, which is ~0.3 Torr at -30°C (the warmest temperature we had planned to operate at). It is ~0.1 Torr at -40°C (the lowest temperature we actually worked at). As He was also necessary to transport HCl, 1 Torr total pressure was regarded as a good compromise.

As the referee correctly points out, we cannot rule out that our uptake coefficients are, to some extent, limited (i.e. reduced) by transport to the surface. Our observation that the uptake coefficient scales linearly with the velocity of ice growth suggests however that this effect is not large, otherwise such plots would have been non-linear. The complex geometry of the reactor precludes accurate assessment and correction of gradients in HCl that may ensue close to the surface at larger values of the uptake coefficient. We now write:

All experiments reported here were conducted at a total pressure of 1.33 mbar (mainly He) and at ice surface temperatures between 194 and 227 K. The pressure in the reactor was chosen as a compromise between detection sensitivity (related to the differential pumping of the MS and reaction time in the ion-molecule reactor of the mass spectrometer, see below) and the need to stay above the vapour pressure of ice (~0.1 Torr at -40°C) whilst avoiding severe limitations of mass-transport to the ice surface at higher pressures (see below).

And…

A linear relationship between $\gamma_{trap}$ and $V_{IG}$ was observed for values of $\gamma_{trap}$ up to ~ 0.02. At larger ice growth velocities some datasets indicated a fall-off in the slope of $\gamma_{trap}$ versus $V_{IG}$, the cause of which may have been a transport related limitation to uptake of HCl to the ice, whereby gradients in the HCl concentration close to the ice-surface result in a reduction in mass-transfer of HCl to the ice. As no simple scheme for correction of the diffusive limitation to uptake exists for the complex geometry of the reactor, values of $\gamma_{trap}$ greater than 0.018 (10% less that the maximum value, 0.02, at which linear behaviour was still observed) were neglected in the final analysis.
* * *
I will be up-front and say that I have not worked through the Karcher trapping model in depth. But my impression is that it is based on a Langmuir adsorption isotherm, as a starting point. If so, to what degree does dissolution in the ice crystals themselves play a role in the uptake, both in the model and in the experimental data? Are thermodynamically stable solutions forming (see work by Domine), or are metastable mixtures forming? If metastable, how long would it take for the enhanced "dissolved" HCl to diffuse out, under non-growth conditions?

This is correct, both analyses are based on Langmuir-type adsorption as the initial step in the trapping process. Our observations, based on gas-phase changes in concentration, do not allow us to draw detailed conclusions regarding the mechanism of incorporation into the growing ice film. The concept of HCl "dissolution" is not treated explicitly, but, as we state, a solubility limit is part of the empirical solution of the Kärcher model. In our work, the partial pressures of HCl and temperatures used, ensure that we are always in the HCl/ice stability region of the phase-diagram, hence we observed no irreversible uptake of HCl to the non-growing ice surface. We now mention this in the experimental section:

The HCl concentrations were kept low to ensure that all experiments were conducted in the HCl-ice stability regions of the phase-diagram (Thibert and Dominé, 1997). In accordance, irreversible uptake (e.g. due to formation of stable hydrates of HCl) was never observed when the ice-growth rate was zero.
* * *
Also, is there uptake along the grain boundaries that may affect the (model or experimental) results?

As we do not use ice single-crystals, we expect that there are grain boundaries but cannot quantify this nor the role of HCl uptake to grain boundaries during ice growth. We have modified to the text to explain this:

Although we attempted to make the ice as reproducible as possible, some scatter in $\gamma_{trap}$ was observed for different ice surface even under apparently identical conditions. This phenomenon has often been observed in studies of trace gas interaction with ice surfaces (Crowley et al., 2010) and may be related to the (variable) presence of grain-boundaries in the ice substrate.

It would certainly be of interest to repeat our experiments using single-crystals and we mention this in the conclusions:

Experiments using ice single-crystals would we useful to constrain the role (if any) of HCl uptake to grain-boundaries on our polycrystalline ice substrate.
* * *
How pure is the water that is condensing? Is there any potential for trace ammonia, for example, to be interacting with the sorbed HCl, so enhancing the uptake?

The (bottled) distilled water we use is HPLC grade. We did not test for the presence of basic gases in the vapour above the water samples we used but do not consider the presence of ammonia in the water likely.
* * *
Not much experimental work has been done on uptake of gases to growing ice in the last few years, and the quality of the earlier studies is not nearly as high as in this work. Nevertheless, I suggest that a brief summary of past studies, upon which this work is somewhat based, could be presented in the Introduction.

The limited experimental work is discussed in detail in the results and discussion section. We now add the references to these studies to the introduction, mentioning that they do not cover the parameter space needed to test the Kärcher trapping model:

The limited experimental work (Diehl et al., 1995; Santachiara et al., 1995; Dominé and Rauzy, 2004; Huthwelker et al., 2004) that is available on the uptake of HCl to growing ice surfaces has been conducted under conditions that do not allow evaluation of the trapping model. These studies are discussed in detail in Section 3.1.1

---

## Referee Comment (RC2) · J. Paul Devlin (Referee) · 27 Aug 2019

This appears to me to be an excellent manuscript in all respects. I view it as most valuable as a quiet challenge to others to be more determined to present quantitative results where possible. The authors have met that challenge for a difficult case and I am moved to offer praise particularly because the paper is fully descriptive of what they have done and how/why the quantitative character places it above so many of the the otherwise comparable studies. Of course quantitative results are not always possible.

I offer no real criticism and have no basis for helpful suggestions. I would suggest that within the first paragraph of page 6, the numerical values of 0.2 and 0.018, differing by

an order of magnitude, do not seem to fit as used.

---

## Author Comment (AC2) · 28 Aug 2019

**Referee 2**

In the following, the referee's comments are reproduced (black) along with our replies (blue) and changes made to the text (red) in the revised manuscript.

**General Comments:**

This appears to me to be an excellent manuscript in all respects. I view it as most valuable as a quiet challenge to others to be more determined to present quantitative results where possible. The authors have met that challenge for a difficult case and I am moved to offer praise particularly because the paper is fully descriptive of what they have done and how/why the quantitative character places it above so many of the the otherwise comparable studies. Of course quantitative results are not always possible.

We thank J. Paul Devlin for this very positive assessment of our manuscript.
* * *
**Specific Comments:**

I would suggest that within the first paragraph of page 6, the numerical values of 0.2 and 0.018, differing by an order of magnitude, do not seem to fit as used.

This was a typo: The value 0.2 should have been 0.02. We now write:

A linear relationship between $\gamma_{trap}$ and $V_{IG}$ was observed for values of $\gamma_{trap}$ up to ~ 0.02. At larger ice growth velocities some datasets indicated a fall-off in the slope of $\gamma_{trap}$ versus $V_{IG}$, the cause of which may have been a transport related limitation to uptake of HCl to the ice, whereby gradients in the HCl concentration close to the ice-surface result in a reduction in mass-transfer of HCl to the ice. As no simple scheme for correction of the diffusive limitation to uptake exists for the complex geometry of the reactor, values of $\gamma_{trap}$ greater than 0.018 (10% less that the maximum value (0.02) at which linear behaviour was still observed) were neglected in the final analysis.